# On the Landscape of Sparse Linear Networks

## Abstract

Network pruning, or sparse network has a long history and practical significance in modern applications. Although the loss functions of neural networks may yield bad landscape due to non-convexity, we focus on linear activation which already owes benign landscape. With no unrealistic assumption, we conclude the following statements for the squared loss objective of general sparse linear neural networks: 1) every local minimum is a global minimum for scalar output with any sparse structure, or non-intersected sparse first layer and dense other layers with orthogonal training data; 2) sparse linear networks have sub-optimal local-min for only sparse first layer due to low rank constraint, or output larger than three dimensions due to the global minimum of a sub-network. Overall, sparsity breaks the normal structure, cutting out the decreasing path in original fully-connected networks.

## 1 Introduction

Deep neural networks (DNNs) have achieved remarkable empirical successes in the domains of computer vision, speech recognition, and natural language processing, sparking great interests in the theory behind their architectures and training. However, DNNs are often found to be highly overparameterized, making them computationally expensive with large amounts of memory and computational power. For example, it may take up to weeks on a modern multi-GPU server for large datasets such as ImageNet (Deng et al., 2009). Hence, DNNs are often unsuitable for smaller devices like embedded electronics, and there is a pressing demand for techniques to optimize models with reduced model size, faster inference and lower power consumption.

Sparse networks, that is, neural networks in which a large subset of the model parameters are zero, have emerged as one of the leading approaches for reducing model parameter count. It has been shown empirically that deep neural networks can achieve state-of-the-art results under high levels of sparsity (Han et al., 2015b; Gale et al., 2019; Louizos et al., 2017a). Modern sparse networks are mainly obtained from network pruning (Zhu & Gupta, 2017; Lee et al., 2018; Liu et al., 2018; Frankle & Carbin, 2018), which has been the subject of a great deal of work in recent years. However, training a sparse network with fixed sparsity patterns is difficult (Evci et al., 2019) and few theoretical understanding of general sparse networks has been provided.

Previous work has already analyze deep neural networks, showing that the non-convexity of the associated loss functions may cause complicated and strange optimization landscapes. However, the property of general sparse networks is poorly understood. Saxe et al. (2013) empirically showed that the optimization of deep linear models exhibits similar properties as deep nonlinear models, and for theoretical development, it is natural to begin with linear models before studying nonlinear models (Baldi & Lu, 2012). In addition, several works (Sun et al., 2020) have show bad minimum exists with nonlinear activation. Hence, it is natural to begin with linear activation to understand the impact of sparsity.

In this article, we go further to consider the global landscape of general sparse linear neural networks. We need to emphasize that dense deep linear networks already satisfy that every local minimum is a global minimum under mild conditions (Kawaguchi, 2016; Lu & Kawaguchi, 2017), but findings are different and complicated for sparse linear network. The goal of this paper is to study the relation between sparsity and local minima with the following contributions:

- First, we point out that every local minimum is a global minimum in scalar target case with any depths, any widths and any sparse structure. Besides, we also briefly show that

      similar results hold for non-overlapping filters and orthogonal data feature when sparsity only occurs in the first layer.

- Second, we find out that sparse connections would already give sub-optimal local minima in general non-scalar case through analytic and numerical examples built on convergence analyze. The local-min may be produced from two situations: a sub-sparse linear network which owes its minimum as a local-min of the original sparse network; a rank-deficient solution between different data features due to sparse connections, while both cases verify the fact that sparsity cuts out the decreasing path in original fully-connected networks.

Overall, we hope our work contributes to a better understanding of the landscape of sparsity network on simple neural networks, and provide insights for future research.

The remainder of our paper is organized as follows. In Section 2, we derive the positive findings of shallow sparse linear networks, providing similar landscape as dense linear networks. In Section 3, we give several examples to show the existence of bad local-min for non-scalar case. In section 4, we briefly generalize the results from shallow to deep sparse linear networks. Some proofs are in Appendix.

## 1.1 RELATED WORK

There is a rapidly increasing literature on analyzing the loss surface of neural network objectives, surveying all of which is well outside our scope. Thus, we only briefly survey the works most related to ours.

**Local minima is Global.** The landscape of a linear network date back to Baldi & Hornik (1989), proving that shallow linear neural networks do not suffer from bad local minima. Kawaguchi (2016) generalized same results to deep linear neural networks, and subsequent several works (Arora et al., 2018; Du & Hu, 2019; Eftekhari, 2020) give direct algorithm-type convergence based on this benign property, though algorithm analysis is beyond the scope of this paper. However, situations are quite complicated with nonlinear activations. Multiple works (Ge et al., 2017; Safran & Shamir, 2018; Yun et al., 2018) show that spurious local minima can happen even in two-layer network with population or empirical loss, some are specific to two-layer and difficult to generalize to general multilayer cases. Another line of works (Arora et al., 2018; Allen-Zhu et al., 2018; Du & Hu, 2019; Du et al., 2018; Li et al., 2018; Mei et al., 2018) understands the landscape of neural network in an overparameterized setting, discovering benign landscape with or without gradient method. Since modern sparse networks reserve few parameters compared to overparameterization, we still seek a fundamental view of sparsity in contrast. Our standpoint is that spurious local minima can happen when applied with specific sparsity even in linear networks.

**Sparse networks.** Sparse networks (Han et al., 2015b;a; Zhu & Gupta, 2017; Frankle & Carbin, 2018; Liu et al., 2018) have a long history, but appears heavily on the experiments, and mainly related to network pruning, which has practical importance for reducing model parameter count and deploying diverse devices. However, training sparse networks (from scratch) suffers great difficulty. Frankle & Carbin (2018) recommend reusing the sparsity pattern found through pruning and train a sparse network from the same initialization as the original training ('lottery') to obtain comparable performance and avoid bad solution. Besides, for fixed sparsity patterns, Evci et al. (2019) attempt to find a decreasing objective path from 'bad' solutions to the 'good' ones in the sparse subspace but fail, showing bad local minima can be produced by pruning, while we give more direct view of simple examples to verify this. Moreover, several recent works also give abundant methods (Molchanov et al., 2017; Louizos et al., 2017b; Lee et al., 2018; Carreira-Perpinán & Idelbayev, 2018) for choosing weights or sparse network structure while achieving similar performance. In theoretical view, Malach et al. (2020) prove that a sufficiently over-parameterized neural network with random weights contains a subnetwork with roughly the same accuracy as the target network, providing guarantee for 'good' sparse networks. Some works analyze convolutional network (Shalev-Shwartz et al., 2017; Du et al., 2018) as a specific sparse structure. Brutzkus & Globerson (2017) analyze non-overlapping and overlapping structure as we do, but with weight sharing to simulate CNN-type structure, and under teacher-student setting with population risk. We do not follow CNN-type network but in general sparse networks, though still linear, to conclude straightforward results.

## 2 LANDSCAPE OF SHALLOW SPARSE LINEAR NETWORKS

### 2.1 PRELIMINARIES AND NOTATION

We use bold-faced letters (e.g., $\mathbf{w}$ and $\boldsymbol{a}$) to denote vectors, capital letters (e.g., $W = [w_{ij}]$ and $A = [a_{ij}]$) for matrices. Let $\mathcal{P}_X$ be the orthogonal projection to the column space of the matrix $X$, and $\lambda_i(H)$ is the $i$-th smallest eigenvalue of a real symmetric matrix $H$.

We consider the training samples and their outputs as $\{(\mathbf{x}_i, \mathbf{y}_i)\}_{i=1}^n \subset \mathbb{R}^{d_x} \times \mathbb{R}^{d_y}$, which may come from unknown distribution $\mathcal{D}$. We form the data matrices $X = [\mathbf{x}_1, \ldots, \mathbf{x}_n]^T \in \mathbb{R}^{n \times d_x}$ and $Y = [\mathbf{y}_1, \ldots, \mathbf{y}_n]^T \in \mathbb{R}^{n \times d_y}$, respectively. In our analysis in Sections 2 and 3, we consider a two-layer (sparse) linear neural network with squared loss:

$$\min_{W,A} L(W, A) := \frac{1}{2}\|Y - XWA\|_F^2, \tag{1}$$

where the first layer weight matrix $W = [\mathbf{w}_1, \ldots, \mathbf{w}_d] \in \mathbb{R}^{d_x \times d}$, and the second layer weight matrix $A = [\boldsymbol{a}_1, \ldots, \boldsymbol{a}_d]^T \in \mathbb{R}^{d \times d_y}$. After weights pruning or sparsity constraint, many weights parameters become zero and would not be updated during retraining. We adopt $\mathcal{S}_j := \{k : w_{kj} = 0\}$ as pruned dimensions in the $j$-th column of $W$, and $-\mathcal{S}_j := \mathcal{S}_j^c = [d_x]\backslash\mathcal{S}_j$, where $[d] := \{1, \ldots, d\}$. In addition, $\mathbf{w}_{j,\mathcal{S}}$ denotes the sub-vector of $\mathbf{w}_j$ choosing the positions in $\mathcal{S}$, $X_{\mathcal{S}}$ the sub-matrix of $X$ choosing the column indices in $\mathcal{S}$.

We let $p_j = d_x - |\mathcal{S}_j|$, where $|\mathcal{S}|$ is the cardinality of the set $\mathcal{S}$. Then $\mathbf{w}_{j,-\mathcal{S}_j} \in \mathbb{R}^{p_j}$ is the remaining $j$-th column in first layer weight which leaves out pruned dimension set $\mathcal{S}_j$. Similarly, $X_{-\mathcal{S}_j} \in \mathbb{R}^{n \times p_j}$ means the remaining data matrix connected to $j$-th node in the first layer.

Finally, for simplicity, we denote $X_{-j} = X_{-\mathcal{S}_j}$, $\mathbf{w}_{-j} = \mathbf{w}_{j,-\mathcal{S}_j}$, and $\widetilde{(\cdot)}$ as the pruned layer weight with several zero elements **not** updated all along, if no ambiguity.

Before we begin, a small note on the sparse structure we concern: there may have unnecessary connections and nodes, such as a node with zero out-degree which can be retrieved and excluded from the final layer to the first layer, and other cases are showing in Appendix C. Thus we do not consider them in the subsequent proof and assume each data dimension has valid output connection, i.e., $\cap_{j=1}^d \mathcal{S}_j = \emptyset$.

### 2.2 SCALAR CASE

In the scalar case, assume $d_y = 1$. We then simplify $A = (a_1, \ldots, a_d)^T$. When pruning any weight $a_i$ in the second layer, the output of the $i$-th node in the first layer contribute zero to final output. Hence $\mathbf{w}_i$ can also be pruned. Without loss of generality, we assume second layer parameters are not pruned. After pruning several parameters, the original problem becomes

$$\min_{\mathbf{w}_{-i}, a_i} L(\widetilde{W}, A) := \frac{1}{2}\left\|Y - (X_{-1}, \ldots, X_{-i}, \ldots, X_{-d})\begin{pmatrix} a_1\mathbf{w}_{-1} \\ \vdots \\ a_d\mathbf{w}_{-d} \end{pmatrix}\right\|_F^2. \tag{2}$$

**Theorem 1** *For a two-layer linear neural network with scalar output and any sparse structure, every local minimum is a global minimum.*

Proof: From Eq. (2), if a local minimizer satisfies $a_i = 0$ for some $1 \le i \le d$, then based on the second order condition for a local minima, we have

$$\begin{pmatrix} \dfrac{\partial^2 L}{\partial a_i^2} & \dfrac{\partial^2 L}{\partial a_i \partial \mathbf{w}_{-i}^T} \\ \dfrac{\partial L}{\partial \mathbf{w}_{-i} \partial a_i} & \dfrac{\partial L}{\partial \mathbf{w}_{-i} \partial \mathbf{w}_{-i}^T} \end{pmatrix} \succeq \mathbf{0}, \tag{3}$$

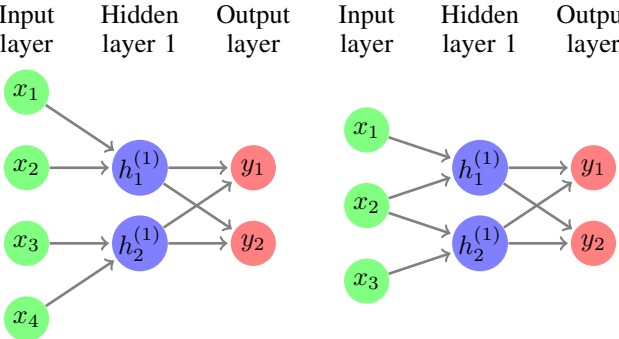

Figure 1: Sparse network without (left) / with (right) overlapping filters in the first layer.

which implies that

$$
\begin{pmatrix}
\mathbf{w}_{-i}^T X_{-i} X_{-i}^T \mathbf{w}_{-i} & -\left(Y - \sum_{i=1}^d X_{-i}\mathbf{w}_{-i}a_i\right)^T X_{-i} \\
-X_{-i}^T\left(Y - \sum_{i=1}^d X_{-i}\mathbf{w}_{-i}a_i\right) & \mathbf{0}
\end{pmatrix} \succeq \mathbf{0}. \tag{4}
$$

Then $X_{-i}^T\left(Y - \sum_{i=1}^d X_{-i}\mathbf{w}_{-i}a_i\right) = \mathbf{0}$, which is the global minimizer condition of $\mathbf{w}_{-i}a_i$.

Otherwise, $a_i \neq 0$, then from the first-order condition for a local minima,

$$
\frac{\partial L}{\partial \mathbf{w}_{-i}} = a_i X_{-i}^T\left(Y - \sum_{i=1}^d X_{-i}\mathbf{w}_{-i}a_i\right) = \mathbf{0},
$$

showing that $X_{-i}^T\left(Y - \sum_{i=1}^d X_{-i}\mathbf{w}_{-i}a_i\right) = \mathbf{0}$, which also gives the global minimizer condition of $\mathbf{w}_{-i}a_i$. Hence every local minimum is a global minimum. □

### 2.3 Non-scalar Case with Dense Second Layer

Now we discuss the case of non-scalar outputs. By the intractable and various sparse structure, we first consider pruning only the first layer while retaining the dense second layer. Then the remaining problem is formulated as follows:

$$
\min_{\mathbf{w}_{-i},\boldsymbol{a}_i} \ L(\widetilde{W}, A) := \frac{1}{2}\left\| Y - \sum_{i=1}^d X_{-i}\mathbf{w}_{-i}\boldsymbol{a}_i^T \right\|_F^2. \tag{5}
$$

Intuitively, if we can separate the weight parameters into $d$ parts, based on linear network results, we can still guarantee no bad local-min. We show that non-overlapping first layer weight or disjoint feature extractor, as the left graph of Figure 1 depicts, and orthogonal data feature meet requirements.

**Theorem 2** *For a two-layer sparse linear neural network with dense second layer, assume that $X$ is full column rank, and $\forall\, i \neq j,\ X_{-i}^T X_{-j} = \mathbf{0}$. Then every local minimum is global.*

Proof: Since $\forall\, i \neq j,\ X_{-i}^T X_{-j} = \mathbf{0}$ and $X$ is full column rank, $X_{-i}$ and $X_{-j}$ share no same columns. Additionally, from our assumption $\cap_{j=1}^d \mathcal{S}_j = \emptyset$, we have $\cap_{j=1}^d \mathcal{S}_j = [d_x]$, meaning that $(X_{-1}, \ldots, X_{-d})$ is $X$ with different arrangement of columns. Hence,

$$
Y = \mathcal{P}_X Y + (I - \mathcal{P}_X)Y = X(X^T X)^{-1}X^T Y + (I - \mathcal{P}_X)Y = \sum_{i=1}^d X_{-i}Z_i + (I - \mathcal{P}_X)Y, \tag{6}
$$

where $Z_i = \left((X^T X)^{-1}X^T Y\right)_{-\mathcal{S}_i}$ is the sub-matrix choosing the row indices in $-\mathcal{S}_i$. Then we only need to consider the objective:

$$
\min_{\mathbf{w}_{-i},\boldsymbol{a}_i} \ L(\widetilde{W}, A) = \frac{1}{2}\left\| \sum_{i=1}^d X_{-i}\left(Z_i - \mathbf{w}_{-i}\boldsymbol{a}_i^T\right) \right\|_F^2 = \frac{1}{2}\sum_{i=1}^d \left\| X_{-i}\left(Z_i - \mathbf{w}_{-i}\boldsymbol{a}_i^T\right) \right\|_F^2. \tag{7}
$$

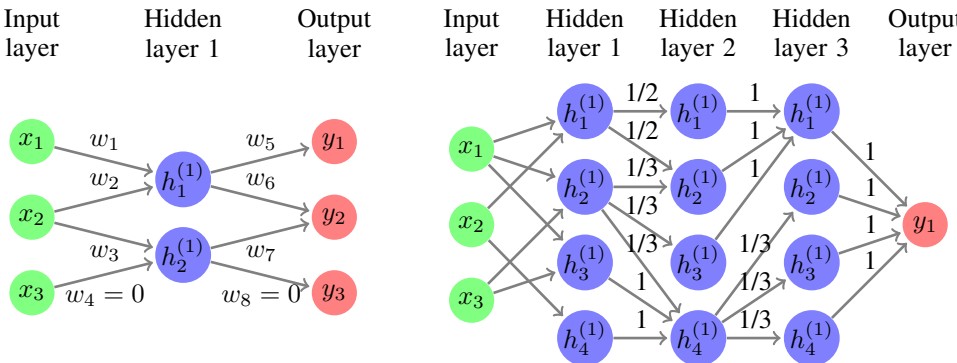

Figure 2: Left: spurious local minimum exists in sparse linear network with $d_y = 3$ shown in above, that is the global minimum of a sub-network. Right: one simple weight assignment for obtaining the global minimum in deep sparse linear network with scalar output.

We will see that the objective has already been separated into $d$ parts, while each part is a two-layer dense linear network with full column rank data matrix. Based on Theorem 2.2 in Lu & Kawaguchi (2017) or Eckart-Young-Mirsky theorem (Eckart & Young, 1936; Mirsky, 1960), we obtain that every local minimum is a global minimum. □

Additionally, we need to explain the assumption that non-overlapping filters in the first layer involves convolution networks (Brutzkus & Globerson, 2017) if weight sharing is used. Otherwise, we will show a bad local minima exists when the first layer is overlapped or training data is not orthogonal in Section 3.

## 2.4 GENERAL CASE WITH $d_y = 2$

Previous findings imply positive results with one-dimension outputs, or specific sparse structure and data assumption. In this subsection, we discuss an arbitrary sparse structure with outputs of dimension $d_y = 2$. We first prove that some connections still owe common benign landscape which can be removed.

**Theorem 3** *For a sparse two-layer network, a node with full out-degree and one in-degree can be removed if we consider the remaining structure with objective under some projected data, having no influence for spurious local minima.*

Previous result simplifies the sparse structure including such a hidden node with one connection to input and full connection to output. Next, we will provide another type reduction with only one connection to output when sparsity is applied to both layers with $d_y = 2$.

We mention the final layer output as Node 1 and Node 2, and the hidden nodes which have only one connection to the output layer as $\mathcal{R}_1$ and $\mathcal{R}_2$ while the remaining full connection set as $\mathcal{R}$. Set $\mathcal{T}_1 = \cap_{j \in \mathcal{R}_1} \mathcal{S}_j, \mathcal{T}_2 = \cap_{j \in \mathcal{R}_2} \mathcal{S}_j$. We define $\mathcal{U}(\mathcal{T}) = \{j : w_{ij} \neq 0, i \in \mathcal{T}\}$ as the hidden node set connected with data feature in $\mathcal{T}$. When the hidden node sets connected to $\mathcal{T}_1$ and $\mathcal{T}_2$ satisfy the condition below, we are able to simply the sparse structure into the dense layer case.

**Theorem 4** *For a sparse two-layer linear network with $d_y = 2$, if $\mathcal{U}(\mathcal{T}_1 \cap \mathcal{T}_2) \cap \mathcal{U}(\mathcal{T}_1 \setminus \mathcal{T}_2) = \emptyset$ and $\mathcal{U}(\mathcal{T}_1 \cap \mathcal{T}_2) \cap \mathcal{U}(\mathcal{T}_2 \setminus \mathcal{T}_1) = \emptyset$, then there is a sub-network with dense second layer optimized with some projected training data, sharing the same local minimum for the remaining parameters.*

The formal proof of the theorem can be found in Appendix E. Additionally, from the proof of Theorem 4, the objective is converted into two objectives with weight sharing in the first layer even the assumption does not meet. Weight sharing structure has been shown in some related work (Shalev-Shwartz et al., 2017; Brutzkus & Globerson, 2017), so we do not give detailed description and leave it as future work.

Now for a sparse two-layer linear network with $d_y = 2$, we focus on the case which has dense second layer. If one hidden node only has one in-degree, then based on Theorem 3, we can remove such node and consider the objective optimized with some projected data. Therefore, each hidden node should have at least two in-degree. Because one hidden node obviously leads to no bad local minima, the least sparse structure has two hidden nodes with totally eight connections (e.g., two constructions in Figure 1). We will show the existence of bad local-minima in Section 3.

## 2.5 General Case with $d_y \geq 3$

We finish this section by discovering that a sub-network with its global minima might yield a spurious local minima of the original sparse network when $d_y \geq 3$.

**Theorem 5** *There exists a spurious local minima that is a global minimum of sub-network in two-layer sparse network when output dimension $d_y \geq 3$.*

Proof: We consider the sparse structure in Figure 2 with only eight connections. The objective is

$$\min_{w_i} \ L(w_1, \ldots, w_8) := \frac{1}{2} \left\| Y - X \begin{pmatrix} w_1 & 0 \\ w_2 & w_3 \\ 0 & w_4 \end{pmatrix} \begin{pmatrix} w_5 & w_6 & 0 \\ 0 & w_7 & w_8 \end{pmatrix} \right\|_F^2$$

$$= \frac{1}{2} \left\| Y - X \begin{pmatrix} w_1 w_5 & w_1 w_6 & 0 \\ w_2 w_5 & w_2 w_6 + w_3 w_7 & w_3 w_8 \\ 0 & w_4 w_7 & w_4 w_8 \end{pmatrix} \right\|_F^2.$$

Let $X = I_3$ and $Y = \begin{pmatrix} 1 & 2 & 0 \\ 2 & 10 & 0 \\ 0 & 0 & 4 \end{pmatrix}$. Clearly, $X$ and $Y$ have full column rank that is the common

assumption in previous work. Then $\begin{pmatrix} w_1 & 0 \\ w_2 & w_3 \\ 0 & w_4 \end{pmatrix} = \begin{pmatrix} 1 & 0 \\ 2 & 2 \\ 0 & 0 \end{pmatrix}, \begin{pmatrix} w_5 & w_6 & 0 \\ 0 & w_7 & w_8 \end{pmatrix} = \begin{pmatrix} 1 & 2 & 0 \\ 0 & 3 & 0 \end{pmatrix}$

satisfy $\nabla L = \mathbf{0}$, and $L(w_1, \ldots, w_8) = 8$. In addition, for any small disturbances $\delta_i, i = 1 \ldots, 8$,

$$\begin{aligned} 2L(w_1 + \delta_1, \ldots, w_8 + \delta_8) &= (\delta_1 + \delta_5 + \delta_1 \delta_5)^2 + (2\delta_1 + \delta_6 + \delta_1 \delta_6)^2 + (\delta_2 + 2\delta_5 + \delta_2 \delta_5)^2 \\ &\quad + (2\delta_2 + 2\delta_6 + \delta_2 \delta_6 + 3\delta_3 + 2\delta_7 + \delta_3 \delta_7)^2 \\ &\quad + (2 + \delta_3)^2 \delta_8^2 + (3 + \delta_7)^2 \delta_4^2 + (\delta_4 \delta_8 - 4)^2 \\ &\geq (2 + \delta_3)^2 \delta_8^2 + (3 + \delta_7)^2 \delta_4^2 + (\delta_4 \delta_8 - 4)^2 \\ &\geq 2 \left[ (2 + \delta_3)(3 + \delta_7) - 4 \right] |\delta_4 \delta_8| + 16. \end{aligned}$$

Since the perturbations $\delta_i$ are small, we have $(2 + \delta_3)(3 + \delta_7) - 4 > 0$. Hence, $L(w_1 + \delta_1, \ldots, w_8 + \delta_8) \geq 8$, verifying the local minimizer.

However, when $\begin{pmatrix} w_1 & 0 \\ w_2 & w_3 \\ 0 & w_4 \end{pmatrix} = \begin{pmatrix} \sqrt{10}/5 & 0 \\ \sqrt{10} & 0 \\ 0 & 2 \end{pmatrix}, \begin{pmatrix} w_5 & w_6 & 0 \\ 0 & w_7 & w_8 \end{pmatrix} = \begin{pmatrix} \sqrt{10}/5 & \sqrt{10} & 0 \\ 0 & 0 & 2 \end{pmatrix}$,

$L(w_1, \ldots, w_8) = 0.18 < 8$. Hence, a bad local minimum exists. $\qquad\square$

We underline that the bad local minimum is produced from the sub-network when $w_4 = w_8 = 0$. Since we encounter no bad local minimum in a dense linear network, sparse connections indeed destroy the benign landscape because sparsity obstructs the decreasing path as Evci et al. (2019) mentioned from experiments.

## 3 Bad Local Minimum with Sparse First Layer

Now we turn back to the dense second layer case in Section 2.3 with $d_y = 2$, and assume $X$ has full column rank. We give an algorithm to check the existence of spurious local minima when $\exists i \neq j$, s.t., $X_{-i}^T X_{-j} \neq \mathbf{0}$.

---

**Algorithm 1 Sparse-2-Opt** $(Z_1, Z_2, D)$: Obtain the solution of two-layer sparse linear neural network with two hidden neurons.

1: **Input:** Target matrix $Z_1, Z_2$ and covariance diagonal matrix $D$.
2: Initialize $\mathbf{w}_2, d_2, \boldsymbol{a}_2$;
3: **while** *not converge* **do**
4:     $\mathbf{w}_1, d_1, \boldsymbol{a}_1 = SVD(Z_1 + D(Z_2 - d_2 \mathbf{w}_2 \boldsymbol{a}_2^T))$;
5:     $\mathbf{w}_2, d_2, \boldsymbol{a}_2 = SVD(Z_2 + D^T(Z_1 - d_1 \mathbf{w}_1 \boldsymbol{a}_1^T))$;
6: **end while**
7: $\mathbf{w}_1 = d_1 \mathbf{w}_1, \mathbf{w}_2 = d_2 \mathbf{w}_2$.
8: **if** $\lambda_1(\nabla^2 L), \lambda_2(\nabla^2 L) \approx 0, \lambda_3(\nabla^2 L) > 0$ **then**
9:     Return the solution $\mathbf{w}_1, \boldsymbol{a}_1, \mathbf{w}_2, \boldsymbol{a}_2$.
10: **else**
11:     Try again from another initialization.
12: **end if**
13: **Output:** $\mathbf{w}_1, \boldsymbol{a}_1, \mathbf{w}_2, \boldsymbol{a}_2$.

---

Notice that we have no rank constraint for the $Z_i$ in Eq. (5). Suppose singular value decomposition of $X_{-i}$ as $X_{-i} = U_i D_i V_i^T$ with $U_i \in \mathbb{R}^{n \times p_i}, D_i \in \mathbb{R}^{p_i \times p_i}, V_i \in \mathbb{R}^{p_i \times p_i}$. Since $D_i$ has full rank, we take $D_i V_i Z_i$ and $D_i V_i \mathbf{w}_i$ as new targets and variables. With a slight abuse of notation, then the problem becomes

$$\min_{\widetilde{W}, A} \frac{1}{2} \left\| \sum_{i=1}^{d} U_i \left( Z_i - \mathbf{w}_i \boldsymbol{a}_i^T \right) \right\|_F^2. \tag{8}$$

In the following, we show $d = 2$ is enough to give counter examples. Similarly, using the singular value decomposition of $U_1^T U_2$ as $U_1^T U_2 = \bar{U} \bar{D} \bar{V}^T$ with a rectangle diagonal matrix $D \in \mathbb{R}^{p_1 \times p_2}$. Notice that $U_1, U_2$ are column orthogonal matrices, thus $D_{ii} \leq 1$, and $|\{i : D_{ii} = 1\}|$ equals to the overlapping columns between $X_{-1}$ and $X_{-2}$. Finally, the objective becomes

$$L(\mathbf{w}_1, \mathbf{w}_2, \boldsymbol{a}_1, \boldsymbol{a}_2) = \frac{1}{2} \|Z_1 - \mathbf{w}_1 \boldsymbol{a}_1^T\|_F^2 + \frac{1}{2} \|Z_2 - \mathbf{w}_2 \boldsymbol{a}_2^T\|_F^2 + \text{tr}[(Z_1 - \mathbf{w}_1 \boldsymbol{a}_1^T)^T D (Z_2 - \mathbf{w}_2 \boldsymbol{a}_2^T)].$$

If we fix $\mathbf{w}_2$ and $\boldsymbol{a}_2$, we can see $\mathbf{w}_1$ and $\boldsymbol{a}_1$ are the best rank-1 approximation of $Z_1 + D(Z_2 - \mathbf{w}_2 \boldsymbol{a}_2^T)$, since $\mathbf{w}_1$ and $\boldsymbol{a}_1$ are the solution of

$$\arg\min_{\mathbf{w}_1, \boldsymbol{a}_1} \|Z_1 + D(Z_2 - \mathbf{w}_2 \boldsymbol{a}_2^T) - \mathbf{w}_1 \boldsymbol{a}_1^T\|_F^2.$$

Similarly, $\mathbf{w}_2$ and $\boldsymbol{a}_2$ are the best rank-1 approximation of $Z_2 + D^T(Z_1 - \mathbf{w}_1 \boldsymbol{a}_1^T)$. Empirically, we use alternating update method to find the solution in Algorithm 1 for two blocks, where $SVD(\cdot)$ is a classical method getting the largest singular value and the corresponding singular vectors.

Since each update does not increase the loss, this makes the convergence of sequence $\mathbf{w}_1, \boldsymbol{a}_1, \mathbf{w}_2, \boldsymbol{a}_2$. Once the algorithm converges, the first-order condition is satisfied and two eigenvectors with zero eigenvalue of the Hessian matrix are decided. Moreover, we can also prove that the convergent solution is indeed a local minima (detail see Appendix B). Otherwise, we examine a local minimum using gradient descent or other optimization method started with noise, if necessary.

Based on Algorithm 1, we find several cases with bad local minima including the overlapping case ($\exists i, D_{ii} = 1$). The results are shown in Table 1. We observe distinct gaps between the local minima because our choice of elements in the $Z_i$ is small. In the non-overlapping setting, the algorithm reaches the local min quickly and shows several different examples. As for the overlapping setting, a simple construction is leaving out the repeated feature away with zero items in the $Z_i$, though we also show bad-min applied with the overlapping data feature in Row 3 in Table 1.

It is interesting to note that for $d = 2$, only at most two local minimum are found, and we can easily broaden the alternating update method into general $d$ case in Appendix D, that will also verify similar observation: at most $d$ local minimum produced by a sparse-first-layer network with hidden $d$ nodes, which leaves as future work. Overall, sparsity breaks the original matrix structure, leading to additional low rank constraint in this case, and still cuts out the decreasing path in the original fully-connected network.

Table 1: Examples found by algorithm with spurious local minimum. All experiments run 600 iterations, except last one with 1000 iterations. $\lambda_i := \lambda_i(\nabla^2 L)$.

| $Z_1$ | $Z_2$ | $D$ | $\lambda_3$ | $\lambda_1, \lambda_2$ | $\|\nabla L\|_2$ | Objective $L$ |
|---|---|---|---|---|---|---|
| $\begin{pmatrix} 1 & 1 \\ 1 & 0 \end{pmatrix}$ | $\begin{pmatrix} 1 & 1 \\ 1 & -1 \end{pmatrix}$ | $\begin{pmatrix} 0.5 & 0 \\ 0 & 0.9 \end{pmatrix}$ | $2.1\cdot10^{-1}$
$1.4\cdot10^{-1}$ | $0\sim10^{-14}$
$0\sim10^{-14}$ | $<10^{-14}$
$<10^{-14}$ | 0.5143043518476
0.6781647585271 |
| $\begin{pmatrix} -2 & 0 \\ 0 & -1 \end{pmatrix}$ | $\begin{pmatrix} 0 & 1 \\ -2 & 2 \end{pmatrix}$ | $\begin{pmatrix} 0.8 & 0 \\ 0 & 0.1 \end{pmatrix}$ | $3.4\cdot10^{-1}$
$1.7\cdot10^{-1}$ | $0\sim10^{-14}$
$0\sim10^{-14}$ | $<10^{-14}$
$<10^{-14}$ | 0.5373672988360
0.6805528480352 |
| $\begin{pmatrix} -1 & 0 \\ 1 & 1 \\ -1 & 0 \end{pmatrix}$ | $\begin{pmatrix} 0 & 0 \\ 1 & 1 \\ -2 & 0 \end{pmatrix}$ | $\begin{pmatrix} 1 & 0 & 0 \\ 0 & 0.6 & 0 \\ 0 & 0 & 0.8 \end{pmatrix}$ | $1.7\cdot10^{-1}$

$2.5\cdot10^{-1}$ | $0\sim10^{-14}$

$0\sim10^{-14}$ | $<10^{-14}$

$<10^{-14}$ | 0.8980944246693

0.4712847600704 |

Additionally, a descent algorithm still will diverge to infinity. For instance, the example in Appendix A shows that there is a sequence diverging to infinity while the function values are decreasing and convergent.

## 4 LANDSCAPE OF DEEP SPARSE LINEAR NETWORKS

In this section, we briefly extend Theorems 1 and 2 to deep sparse linear networks and leave the proof in Appendix F. The intuition is that deep linear networks have similar landscape property as the shallow case (Lu & Kawaguchi, 2017; Eftekhari, 2020). However, understanding the landscape of an arbitrary deep sparse linear network is still complicated.

**Theorem 6** *For a deep sparse linear neural network with scalar output ($d_y = 1$) and any sparse structure, every local minimum is a global minimum.*

The proof intuition can be described by induction based on shallow linear networks. The above theorem shows that sparsity introduces no bad-min when applied with scalar target. In addition, we give a simple choice for obtaining a global minimizer below.

**How to obtain a global minimizer in scalar case:** One way is to set the first-layer weights as the global minimizer in the two-layer case with $a_i = 1$, then the remaining layers uniformly distribute the output of each node to the next layer. For example, if one node has $k$ output connections, then each connection assigns weight $1/k$. Hence, the sum of each layer output remains the best solution to approximate target $Y$ (see the right graph of Figure 2 for example).

**Theorem 7** *For a deep sparse linear neural network with a sparse first layer and dense other layers, assume that $X, Y$ have full column rank, and $\forall i \neq j$, $X_{-i}^T X_{-j} = \mathbf{0}$. If $d_i \geq \min\{d_1, d_y\}, \forall i \geq 1$, where $d_i$ is the hidden width in the $i$-th layer, then every local minimum is a global minimum.*

Note that under our assumption $d_i \geq \min\{d_1, d_y\}, \forall i \geq 1$, the deep linear network we study has the same solution as the shallow linear network when the first-layer weight fixed. Hence, the optimal value for our objective function is equal to the optimal value of the shallow network problem.

## 5 DISCUSSION

We have discussed the landscape of sparse linear networks with several arguments. On the positive side, spurious local minimum does not exist when the objective applied with scalar target, or with separated first layer and orthogonal training data. On the negative side, we have discovered the bad local minimum when the previous conditions are violated in a general sparse two-layer linear network, that is, one is generated from low rank constraint, another is produced from sub-sparse structure. Both the cases show that sparsity cuts out the decreasing path in the original fully-connected network. Since dense linear networks possess benign landscape, we have concluded that sparsity or network pruning destroys the favourable solutions. However, some heuristic algorithms combining training and pruning still work well in practice, leading to mystery of modern network pruning methods and sparse network design. Other interesting questions for future research include understanding the gap between global minimum and spurious local minimum, or showing a similar performance of bad-min, particularly, combining with pruning algorithms.

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

## A  Decreasing Path of Sparse Linear network with Sparse First Layer

In addition, there still exists decreasing path to infinity:

$$
\begin{aligned}
\min_{w_i} \ L(w_1, \ldots, w_8) &:= \frac{1}{2} \left\| Y - X \begin{pmatrix} w_1 & 0 \\ w_2 & w_3 \\ 0 & w_4 \end{pmatrix} \begin{pmatrix} w_5 & w_6 \\ w_7 & w_8 \end{pmatrix} \right\|_F^2 \\
&= \frac{1}{2} \left\| Y - X \begin{pmatrix} w_1 w_5 & w_1 w_6 \\ w_2 w_5 + w_3 w_7 & w_2 w_6 + w_3 w_8 \\ w_4 w_7 & w_4 w_8 \end{pmatrix} \right\|_F^2
\end{aligned}
\tag{9}
$$

$X = I_3, Y = \begin{pmatrix} 1 & 0 \\ 0 & 1 \\ 1 & 1 \end{pmatrix}$, Choose $(w_1, w_2, w_3, w_4, w_5, w_6, w_7, w_8) = (-\frac{1}{\sqrt{k}}, -\sqrt{k}, 1, 1, \frac{1}{\sqrt{k}}, 0, 1, 1)$,

with $k \in \mathbb{N}^+$. then $2L(w_1, \ldots, w_8) = (\frac{1}{k} + 1)^2 > 1$ decreases when $k$ increases. Since $\min_{w_i} L(w_1, \ldots, w_8) = 0$, hence we get a decreasing path to infinity, but not a global minimum.

## B  Algorithm Analysis

We built algorithm guarantee in the following:

First, since each update step, the objective doesn't increase, then the algorithm will converge.

Second, we verify that the convergent solution $(\mathbf{w}_1^*, \boldsymbol{a}_1^*, \mathbf{w}_2^*, \boldsymbol{a}_2^*)$ satisfy zero gradient. Recall the first-order condition:

$$
\begin{aligned}
-\frac{\partial L}{\partial \mathbf{w}_1} &= \left( Z_1 - \mathbf{w}_1 \boldsymbol{a}_1^T + D(Z_2 - \mathbf{w}_2 \boldsymbol{a}_2^T) \right) \boldsymbol{a}_1 = \left( Z_1 + D(Z_2 - \mathbf{w}_2 \boldsymbol{a}_2^T) \right) \boldsymbol{a}_1 - \boldsymbol{a}_1^T \boldsymbol{a}_1 \mathbf{w}_1, \\
-\frac{\partial L}{\partial \boldsymbol{a}_1} &= \left( Z_1 - \mathbf{w}_1 \boldsymbol{a}_1^T + D(Z_2 - \mathbf{w}_2 \boldsymbol{a}_2^T) \right)^T \mathbf{w}_1 = \left( Z_1 + D(Z_2 - \mathbf{w}_2 \boldsymbol{a}_2^T) \right)^T \mathbf{w}_1 - \mathbf{w}_1^T \mathbf{w}_1 \boldsymbol{a}_1, \\
-\frac{\partial L}{\partial \mathbf{w}_2} &= \left( Z_2 - \mathbf{w}_2 \boldsymbol{a}_2^T + D^T(Z_1 - \mathbf{w}_1 \boldsymbol{a}_1^T) \right) \boldsymbol{a}_2 = \left( Z_2 + D^T(Z_1 - \mathbf{w}_1 \boldsymbol{a}_1^T) \right) \boldsymbol{a}_2 - \boldsymbol{a}_2^T \boldsymbol{a}_2 \mathbf{w}_2, \\
-\frac{\partial L}{\partial \boldsymbol{a}_2} &= \left( Z_2 - \mathbf{w}_2 \boldsymbol{a}_2^T + D^T(Z_1 - \mathbf{w}_1 \boldsymbol{a}_1^T) \right)^T \mathbf{w}_2 = \left( Z_2 + D^T(Z_1 - \mathbf{w}_1 \boldsymbol{a}_1^T) \right)^T \mathbf{w}_2 - \mathbf{w}_2^T \mathbf{w}_2 \boldsymbol{a}_2.
\end{aligned}
\tag{10}
$$

Notice that $\mathbf{w}_1^*, \boldsymbol{a}_1^*$ is the best rank-1 approximation of $Z_1 + D(Z_2 - \mathbf{w}_2^* \boldsymbol{a}_2^{*T})$, and $\mathbf{w}_2^*, \boldsymbol{a}_2^*$ are the best rank-1 approximation of $Z_2 + D^T(Z_1 - \mathbf{w}_1^* \boldsymbol{a}_1^{*T})$. Then we have already got a solution $(\mathbf{w}_1^*, \boldsymbol{a}_1^*, \mathbf{w}_2^*, \boldsymbol{a}_2^*)$ with zero gradient.

Third, we verify that the convergent solution is a local minimizer through the conditions we checked. Set $\mathbf{r}_1 = Z_1 + D(Z_2 - \mathbf{w}_2 \boldsymbol{a}_2^T)$, $\mathbf{r}_2 = Z_2 + D^T(Z_1 - \mathbf{w}_1 \boldsymbol{a}_1^T)$. Then

$$
\nabla^2 L(\mathbf{w}_1, \boldsymbol{a}_1, \mathbf{w}_2, \boldsymbol{a}_2) = \begin{pmatrix}
\boldsymbol{a}_1^T \boldsymbol{a}_1 I_{p_1} & -\mathbf{r}_1 + 2\mathbf{w}_1 \boldsymbol{a}_1^T & \boldsymbol{a}_1^T \boldsymbol{a}_2 D & D\mathbf{w}_2 \boldsymbol{a}_1^T \\
\left(-\mathbf{r}_1 + 2\mathbf{w}_1 \boldsymbol{a}_1^T\right)^T & \mathbf{w}_1^T \mathbf{w}_1 I_{d_y} & \boldsymbol{a}_2 \mathbf{w}_1^T D & \mathbf{w}_1^T D\mathbf{w}_2 I_{d_y} \\
\boldsymbol{a}_1^T \boldsymbol{a}_2 D^T & D^T \mathbf{w}_1 \boldsymbol{a}_2^T & \boldsymbol{a}_2^T \boldsymbol{a}_2 I_{p_2} & -\mathbf{r}_2 + 2\mathbf{w}_2 \boldsymbol{a}_2^T \\
\boldsymbol{a}_1 \mathbf{w}_2^T D^T & \mathbf{w}_1^T D\mathbf{w}_2 I_{d_y} & \left(-\mathbf{r}_2 + 2\mathbf{w}_2 \boldsymbol{a}_2^T\right)^T & \mathbf{w}_2^T \mathbf{w}_2 I_{d_y}
\end{pmatrix}
\tag{11}
$$

Set $H^* := \nabla^2 L(\mathbf{w}_1^*, \boldsymbol{a}_1^*, \mathbf{w}_2^*, \boldsymbol{a}_2^*)$. Observe that

$$
\left( \mathbf{w}_1^{*T}, -\boldsymbol{a}_1^{*T}, \mathbf{0}^T, \mathbf{0}^T \right) H^* = \mathbf{0}, \ \left( \mathbf{0}^T, \mathbf{0}^T, \mathbf{w}_2^{*T}, -\boldsymbol{a}_2^{*T} \right) H^* = \mathbf{0},
$$

showing that $H^*$ has zero eigenvalue with at least two eigenvectors $\mathbf{v}_1 = \left( \mathbf{w}_1^{*T}, -\boldsymbol{a}_1^{*T}, \mathbf{0}^T, \mathbf{0}^T \right)^T$ and $\mathbf{v}_2 = \left( \mathbf{0}^T, \mathbf{0}^T, \mathbf{w}_2^{*T}, -\boldsymbol{a}_2^{*T} \right)^T$.

Suppose the third smallest eigenvalue is $\lambda_3 \geq \epsilon > 0$, then for any direction $\mathbf{v}$ with $\|\mathbf{v}\|_2 = 1$, we have $\mathbf{v} = \alpha_1 \bar{\mathbf{v}}_1 + \alpha_2 \bar{\mathbf{v}}_2 + \alpha_3 \bar{\mathbf{v}}_3$ with $\mathbf{v}_3 \perp \mathbf{v}_1, \mathbf{v}_3 \perp \mathbf{v}_2, \sum_{i=1}^3 \alpha_i^2 = 1$, and $\bar{\mathbf{w}} := \mathbf{w}/\|\mathbf{w}\|_2$.

If $\alpha_3 \neq 0$, then $\mathbf{v}H\mathbf{v} = \alpha_3^2 \bar{\mathbf{v}}_3 H \bar{\mathbf{v}}_3 \geq \alpha_3^2 \lambda_3 \geq \alpha_3^2 \epsilon \geq 0$. Otherwise, we set $\mathbf{v} = \delta_1 \mathbf{v}_1 + \delta_2 \mathbf{v}_2$ with small $\delta_1, \delta_2$ as perturbation, and the perturbed parameters are notated as $\hat{\mathbf{w}}_1, \tilde{\boldsymbol{a}}_1, \tilde{\mathbf{w}}_2, \tilde{\boldsymbol{a}}_2 = (1-\delta_1)\mathbf{w}_1^*, (1-\delta_2)\mathbf{w}_2^*, (1+\delta_1)\boldsymbol{a}_1^*, (1+\delta_2)\boldsymbol{a}_2^*$, which yields

$$
\begin{aligned}
L(\tilde{\mathbf{w}}_1, \tilde{\boldsymbol{a}}_1, \tilde{\mathbf{w}}_2, \tilde{\boldsymbol{a}}_2) &= \| X_1 \left( Z_1 - (1-\delta_1^2)\mathbf{w}_1 \boldsymbol{a}_1^T \right) + X_2 \left( Z_2 - (1-\delta_2^2)\mathbf{w}_2 \boldsymbol{a}_2^T \right) \|_F^2 \\
&= \| X_1 \left( Z_1 - \mathbf{w}_1 \boldsymbol{a}_1^T \right) + X_2 \left( Z_2 - \mathbf{w}_2 \boldsymbol{a}_2^T \right) \|_F^2 + \| \delta_1^2 \mathbf{w}_1 \boldsymbol{a}_1^T + \delta_2^2 \mathbf{w}_2 \boldsymbol{a}_2^T \|_F^2 \\
&\quad + 2\delta_1^2 \, tr[\left( \mathbf{w}_1 \boldsymbol{a}_1^T \right)^T \left( \mathbf{r}_1 - \mathbf{w}_1 \boldsymbol{a}_1^T \right)] + 2\delta_2^2 \, tr[\left( \mathbf{w}_2 \boldsymbol{a}_2^T \right)^T \left( \mathbf{r}_2 - \mathbf{w}_2 \boldsymbol{a}_2^T \right)] \\
&= \| X_1 \left( Z_1 - \mathbf{w}_1 \boldsymbol{a}_1^T \right) + X_2 \left( Z_2 - \mathbf{w}_2 \boldsymbol{a}_2^T \right) \|_F^2 + \| \delta_1^2 \mathbf{w}_1 \boldsymbol{a}_1^T + \delta_2^2 \mathbf{w}_2 \boldsymbol{a}_2^T \|_F^2 \\
&= L(\mathbf{w}_1, \boldsymbol{a}_1, \mathbf{w}_2, \boldsymbol{a}_2) + \| \delta^2 \alpha_1^2 \mathbf{w}_1 \boldsymbol{a}_1^T + \delta^2 \alpha_1^2 \mathbf{w}_2 \boldsymbol{a}_2^T \|_F^2 \geq L(\mathbf{w}_1, \boldsymbol{a}_1, \mathbf{w}_2, \boldsymbol{a}_2).
\end{aligned}
\tag{12}
$$

Third equality holds for the rank-1 approximation of the solution. Hence, the convergent solution is a local minimizer.

Fourth, due to numerical error, we can not obtain exact convergent solution, but we are able to obtain approximate solution $(\mathbf{w}_1^t, \boldsymbol{a}_1^t, \mathbf{w}_2^t, \boldsymbol{a}_2^t)$ after $t$ iterations with $L(\mathbf{w}_1^t, \boldsymbol{a}_1^t, \mathbf{w}_2^t, \boldsymbol{a}_2^t) - L(\mathbf{w}_1^*, \boldsymbol{a}_1^*, \mathbf{w}_2^*, \boldsymbol{a}_2^*) \leq \epsilon^2$, and then use Weyl's inequality (Safran & Shamir, 2018, Theorem 2),

$$
\left| \lambda_i (\nabla^2 L(\mathbf{w}_1^t, \boldsymbol{a}_1^t, \mathbf{w}_2^t, \boldsymbol{a}_2^t)) - \lambda_i (\nabla^2 L(\mathbf{w}_1^*, \boldsymbol{a}_1^*, \mathbf{w}_2^*, \boldsymbol{a}_2^*)) \right| < O(\epsilon),
$$

where $\lambda_i(H)$ is the $i$-th smallest eigenvalue of the real symmetric matrix $H$. Therefore, if the approximate solution is approximate positive semi-definite with a large third smallest eigenvalue, we conclude the convergent solution is a local minimizer.

## C  USELESS CONNECTIONS AND NODES IN SPARSE NETWORK

In this section, we explain several kinds of unnecessary connections suffered from sparsity or network pruning.

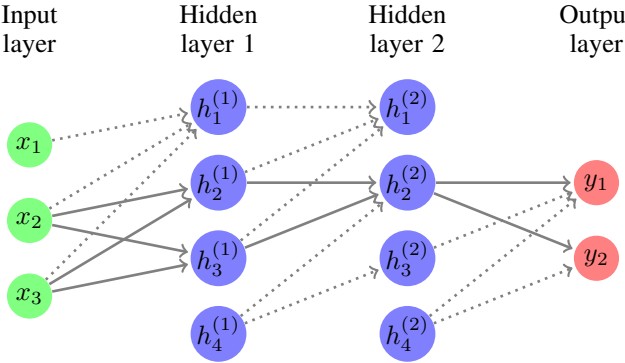

Figure 3: An example of sparse network with no bias. Lines are connections of original sparse network, dotted lines are useless connections that can be removed, and solid lines are effective connections.

1. Zero out-degree I: if a node have zero out-degree, such as $h_1^{(2)}$ in Figure 3, we can eliminate the input connections.

2. Zero out-degree II: if a node have zero out-degree when removed output connections in latter layers, such as $h_1^{(1)}$ in Figure 3. Though it owes one output connection, the connected node $h_1^{(2)}$ is zero out-degree, hence the connection can be removed, leading to zero out-degree. We can eliminate the input connections of $h_1^{(1)}$ as well.

3. Zero in-degree I: if a node have zero in-degree, such as $h_4^{(2)}$ and $h_4^{(1)}$ in Figure 3, we can eliminate the output connections, but notice that when the node has a bias term, then we can not remove output connections since the bias constant will still propagate to subsequent layers.

4. Zero in-degree II: if a node have zero in-degree when removed input connections in former layers, such as $h_3^{(2)}$ in Figure 3. Though it owes one input connection, the connected node $h_4^{(1)}$ is zero in-degree, hence the connection can be removed, leading to zero in-degree. We can eliminate the output connections of $h_3^{(2)}$ as well.

## D  GENERAL $d$ BLOCKS ALGORITHM

---

**Algorithm 2 Sparse-$d$-Opt** $(X_1, \ldots, X_d, Z_1, \ldots, Z_d)$: Obtain the solution of two-layer sparse linear neural network with $d$ hidden neurons.

---
1: **Input:** Input matrix $X_1, \ldots, X_d$. Target matrix $Z_1, \ldots, Z_d$;
2: Initialize $\mathbf{w}_i, d_i, \boldsymbol{a}_i, i = 2, \ldots, d$;
3: **while** $not\ converge$ **do**
4:    **for** $i = 1, \ldots, d$ **do**
5:       $\mathbf{w}_i, d_i, \boldsymbol{a}_i = SVD(Z_i + \sum_{j \neq i} X_i^T X_j (Z_j - d_j \mathbf{w}_j \boldsymbol{a}_j^T))$;
6:    **end for**
7: **end while**
8: $\mathbf{w}_i = d_i \mathbf{w}_i, i = 1, \ldots, d$.
9: **if** $\lambda_1(\nabla^2 L), \ldots, \lambda_d(\nabla^2 L) \approx 0, \lambda_{d+1}(\nabla^2 L) > 0$ **then**
10:    Return the solution $\mathbf{w}_i, \boldsymbol{a}_i, i = 1, \ldots, d$.
11: **else**
12:    Try again from another initialization.
13: **end if**
14: **Output:** $\mathbf{w}_i, \boldsymbol{a}_i, i = 1, \ldots, d$.

---

The analysis that the convergent solution is a local minimizer is similar to $d = 2$, so we are not going to repeat the details. We list some examples searched for $d > 2$ below.

$d = 3$: Target: $Z_1 = \begin{pmatrix} 0 & 0 \\ 0 & 1 \end{pmatrix}, Z_2 = \begin{pmatrix} 1 & 0 \\ 1 & 1 \end{pmatrix}, Z_3 = \begin{pmatrix} 1 & 0 \\ -1 & -1 \end{pmatrix}.$

Training data: $X^T X = \begin{pmatrix} 1.0 & 0.0 & -0.088 & -0.599 & -0.234 & 0.178 \\ 0.0 & 1.0 & -0.163 & 0.429 & -0.529 & 0.431 \\ -0.088 & -0.163 & 1.0 & -0.0 & 0.558 & 0.193 \\ -0.599 & 0.429 & -0.0 & 1.0 & -0.357 & -0.244 \\ -0.234 & -0.529 & 0.558 & -0.357 & 1.0 & -0.0 \\ 0.178 & 0.431 & 0.193 & -0.244 & -0.0 & 1.0 \end{pmatrix}$

Local minimum found:

Table 2: Examples found by algorithm with spurious local minimum when $d_y = 3$. All experiments run 2000 iterations. $\lambda_i := \lambda_i(\nabla^2 L)$.

| $\lambda_{d+1}$ | $\lambda_1, \ldots, \lambda_d$ | $\|\nabla L\|_2$ | Objective $L$ |
|---|---|---|---|
| $1.9 \cdot 10^{-1}$ | $0 \sim 10^{-14}$ | $< 10^{-14}$ | 0.357481957 |
| $9.7 \cdot 10^{-2}$ | $0 \sim 10^{-14}$ | $< 10^{-14}$ | 0.521705675 |
| $4.9 \cdot 10^{-2}$ | $0 \sim 10^{-14}$ | $< 10^{-14}$ | 0.539730382 |

## E  MISSING PROOFS FOR SECTION 2

### E.1  PROOF OF THEOREM 3

Proof: Suppose the $j$-th node has full out-degree and one in-degree, so that $\mathbf{w}_{-j} \in \mathbb{R}$. We treat objective with fixed other weights and only consider optimizing $w_{-j}, \boldsymbol{a}_j$.

$$\min_{w_j, \boldsymbol{a}_j} \ell(w_j, \boldsymbol{a}_j) = \frac{1}{2} \left\| \widetilde{Y} - X_{-j} w_j \boldsymbol{a}_j^T \right\|_F^2, \quad \widetilde{Y} := Y - \sum_{i \neq j} X_{-i} \mathbf{w}_{-i} \boldsymbol{a}_i^T. \tag{13}$$

Based on the proof of scalar case, a local minimizer $(w_j^*, \boldsymbol{a}_j^*)$ of $\ell(w_j, \boldsymbol{a}_j)$ must satisfy the condition $X_{-j}^T \left( \widetilde{Y} - X_{-j} w_j^* \boldsymbol{a}_j^{*T} \right) = 0$, showing that $\ell(w_j^*, \boldsymbol{a}_j^*) = \frac{1}{2} \| \left( I - \mathcal{P}_{X_{-j}} \right) \widetilde{Y} \|_F^2$. Therefore, the objective with remaining weights becomes:

$$\min_{\mathbf{w}_{-i}, \boldsymbol{a}_i, i \neq j} \frac{1}{2} \left\| \left( I_n - \mathcal{P}_{X_{-j}} \right) Y - \sum_{i \neq j} \left( I_n - \mathcal{P}_{X_{-j}} \right) X_{-i} \mathbf{w}_{-i} \boldsymbol{a}_i^T \right\|_F^2. \tag{14}$$

We define $\left( \left( I_n - \mathcal{P}_{X_{-j}} \right) X_{\mathcal{S}_j}, \left( I_n - \mathcal{P}_{X_{-j}} \right) Y \right)$ as new training dataset which is the projection into the orthogonal complement of $X_{-j}$, and remove some elements in $\mathbf{w}_{-i}$ corresponding to the column in $X_{-j}$. Moreover, if $X$ has full column rank, then projected data $\left( I_n - \mathcal{P}_{X_{-j}} \right) X_{\mathcal{S}_j}$ still has full column rank. Hence, removing the above connections doesn't affect the spurious local minima since these connections preserve certain solution. $\qquad \square$

### E.2 PROOF OF THEOREM 4

Proof: The original loss function can be formulated as below,

$$2L(\widetilde{W}, \widetilde{A}) = \left\| Y_1 - \sum_{i \in \mathcal{R}_1} X_{-i} \mathbf{w}_{-i} a_{i1} - \sum_{j \in \mathcal{R}} X_{-j} \mathbf{w}_{-j} a_{j1} \right\|_F^2 + \left\| Y_2 - \sum_{i \in \mathcal{R}_2} X_{-i} \mathbf{w}_{-i} a_{i2} - \sum_{j \in \mathcal{R}} X_{-j} \mathbf{w}_{-j} a_{j2} \right\|_F^2.$$

Under similar analysis as scalar case,

$$\forall i \in \mathcal{R}_1, \ X_{-i}^T \left( Y_1 - \sum_{i \in \mathcal{R}_1} X_{-i} \mathbf{w}_{-i} a_{i1} - \sum_{j \in \mathcal{R}} X_{-j} \mathbf{w}_{-j} a_{j1} \right) = 0.$$

$$\forall i \in \mathcal{R}_2, \ X_{-i}^T \left( Y_2 - \sum_{i \in \mathcal{R}_2} X_{-i} \mathbf{w}_{-i} a_{i2} - \sum_{j \in \mathcal{R}} X_{-j} \mathbf{w}_{-j} a_{j2} \right) = 0.$$

Hence,

$$\begin{pmatrix} \mathbf{w}_{-i_1} a_{i_1 1} \\ \vdots \\ \mathbf{w}_{-i_j} a_{i_j 1} \\ \vdots \\ \mathbf{w}_{-i_{|\mathcal{R}_1|}} a_{i_{|\mathcal{R}_1|} 1} \end{pmatrix} = \left( X_{-i_1}, \cdots, X_{-i_j}, \cdots, X_{-i_{|\mathcal{R}_1|}} \right)^+ \left( Y_1 - \sum_{j \in \mathcal{R}} X_{-j} \mathbf{w}_{-j} a_{j1} \right), \ i_j \in \mathcal{R}_1.$$

$$\begin{pmatrix} \mathbf{w}_{-i_1} a_{i_1 2} \\ \vdots \\ \mathbf{w}_{-i_j} a_{i_j 2} \\ \vdots \\ \mathbf{w}_{-i_{|\mathcal{R}_2|}} a_{i_{|\mathcal{R}_1|} 1} \end{pmatrix} = \left( X_{-i_1}, \cdots, X_{-i_j}, \cdots, X_{-i_{|\mathcal{R}_2|}} \right)^+ \left( Y_2 - \sum_{j \in \mathcal{R}} X_{-j} \mathbf{w}_{-j} a_{j2} \right), \ i_j \in \mathcal{R}_2.$$

Then the objective becomes:

$$\left\| \left( I_n - \mathcal{P}_{X_{-\mathcal{T}_1}} \right) \left( Y_1 - \sum_{j \in \mathcal{R}} X_{-j} \mathbf{w}_{-j} a_{j1} \right) \right\|_2^2 + \left\| \left( I_n - \mathcal{P}_{X_{-\mathcal{T}_2}} \right) \left( Y_2 - \sum_{j \in \mathcal{R}} X_{-j} \mathbf{w}_{-j} a_{j2} \right) \right\|_2^2.$$

We can see the objective is separated into two parts with shared sparse first-layer weights. Notice that if $i \notin \mathcal{T}_1$, then $X_i \in X_{\mathcal{T}_1}$, hence $\left( I_n - P_{X_{-\mathcal{T}_1}} \right) X_i = 0$. Therefore, we simplify the problem as

$$\min_{\widetilde{W}, \widetilde{A}} L(\widetilde{W}, \widetilde{A}) = \frac{1}{2} \left\| \left( I_n - \mathcal{P}_{X_{-\mathcal{T}_1}} \right) \left( Y_1 - \sum_{i \in \mathcal{T}_1} X_i \sum_{j:i \notin \mathcal{S}_j} w_{ij} a_{j1} \right) \right\|_2^2$$

$$+ \frac{1}{2} \left\| \left( I_n - \mathcal{P}_{X_{-\mathcal{T}_2}} \right) \left( Y_2 - \sum_{i \in \mathcal{T}_2} X_i \sum_{j:i \notin \mathcal{S}_j} w_{ij} a_{j2} \right) \right\|_2^2. \tag{15}$$

Use previous analysis again on $\mathcal{T}_1 \setminus \mathcal{T}_2$ in first output dimension and $\mathcal{T}_2 \setminus \mathcal{T}_1$ in second output dimension since no overlap in parameters by the condition $\mathcal{U}(\mathcal{T}_1 \cap \mathcal{T}_2) \cap \mathcal{U}(\mathcal{T}_1 \setminus \mathcal{T}_2) = \emptyset$ and $\mathcal{U}(\mathcal{T}_1 \cap \mathcal{T}_2) \cap \mathcal{U}(\mathcal{T}_2 \setminus \mathcal{T}_1) = \emptyset$. Therefore, we simplify the problem again as

$$
\min_{\widetilde{W}, \widetilde{A}} L(\widetilde{W}, \widetilde{A}) = \frac{1}{2} \left\| \left( I_n - \mathcal{P}_{\left( I_n - \mathcal{P}_{X_{-\mathcal{T}_1}} \right) X_{\mathcal{T}_1 \setminus \mathcal{T}_2}} \right) \left( I_n - \mathcal{P}_{X_{-\mathcal{T}_1}} \right) \left( Y_1 - \sum_{i \in \mathcal{T}_1 \cap \mathcal{T}_2} X_i \sum_{j : i \notin \mathcal{S}_j} w_{ij} a_{j1} \right) \right\|_2^2
$$
$$
+ \frac{1}{2} \left\| \left( I_n - \mathcal{P}_{\left( I_n - \mathcal{P}_{X_{-\mathcal{T}_1}} \right) X_{\mathcal{T}_2 \setminus \mathcal{T}_1}} \right) \left( I_n - \mathcal{P}_{X_{-\mathcal{T}_2}} \right) \left( Y_2 - \sum_{i \in \mathcal{T}_1 \cap \mathcal{T}_2} X_i \sum_{j : i \notin \mathcal{S}_j} w_{ij} a_{j2} \right) \right\|_2^2.
$$

Using the fact that $(I_n - \mathcal{P}_{W_1})(I_n - \mathcal{P}_{W_2}) = I_n - \mathcal{P}_{(W_1, W_2)}$ if $W_1^T W_2 = \mathbf{0}$. Hence the remaining problem is same as

$$
\min_{\widetilde{W}, \widetilde{A}} L(\widetilde{W}, \widetilde{A}) = \frac{1}{2} \sum_{k=1}^{2} \left\| \left( I_n - \mathcal{P}_{X_{-\mathcal{T}_1 \cap \mathcal{T}_2}} \right) \left( Y_k - \sum_{i \in \mathcal{T}_1 \cap \mathcal{T}_2} X_i \sum_{j : i \notin \mathcal{S}_j} w_{ij} a_{jk} \right) \right\|_2^2.
$$

Therefore, the remaining network structure has dense second layer. $\qquad\square$

## F  MISSING PROOFS FOR SECTION 4

The objective of a deep linear network with squared loss is

$$
\min_{W^{(1)}, \ldots, W^{(L)}} \frac{1}{2} \| Y - X W^{(1)} \cdots W^{(L)} \|_F^2, \tag{16}
$$

where the $i$-th layer weight matrix $W^{(i)} \in \mathbb{R}^{d_{i-1} \times d_i}$, $d_0 = d_x$, $d_L = d_y$, Data matrix $X \in \mathbb{R}^{n \times d_x}$, $Y \in \mathbb{R}^{n \times d_y}$. We adopt $\mathcal{S}_j^{(i)} = \{k : W_{kj}^{(i)} = 0\}$ as pruned dimensions in $j$-th column of $W^{(i)}$. Besides, $W_{j,-\mathcal{S}}^{(i)}$ as the remaining $j$-th column in $i$-th layer weight which leaves out pruned dimension set $\mathcal{S}$. For simplification, we denote $\mathbf{w}_{-j}^{(i)} = \mathbf{w}_{j,-\mathcal{S}_j^{(i)}}^{(i)} \in \mathbb{R}^{\left( d_{i-1} - |\mathcal{S}_j^{(i)}| \right)}$, $w_{jk}^{(i)} = W_{jk}^{(i)}$, and $\widetilde{W}^{(i)}$ as the pruned weight matrix with several zero elements as before.

### F.1  PROOF OF THEOREM 6

Proof: Using induction. Base on Theorem 1, we have already proof two layer case. If the result holds for $(L-1)$-layer sparse linear network, we consider $L$ layer case. We denote $X_{new} := X \widetilde{W}^{(1)}$ as new training set, and $\ell := Y - X \widetilde{W}^{(1)} \cdots \widetilde{W}^{(L)}$. Then based on inductive assumption, $\ell^T X_{new} = \mathbf{0}$, showing that

$$
\ell^T X_{-i} \mathbf{w}_{-i}^{(1)} = \mathbf{0}, \ \forall 1 \leq i \leq d_1. \tag{17}
$$

Combined with first-order condition:

$$
\frac{\partial L}{\partial \mathbf{w}_{-i}^{(1)}} = -\ell^T X_{-i} (\widetilde{W}^{(2)} \cdots \widetilde{W}^{(L)})_i = \mathbf{0}.
$$

If $(\widetilde{W}^{(2)} \cdots \widetilde{W}^{(L)})_i \neq 0$, then $\ell^T X_{-i} = 0$, which satisfies the global minimizer condition. Otherwise, any value of $\mathbf{w}_{-i}^{(1)}$ doesn't change the loss since the forward path already contribute zero to the final output. Hence, arbitrary choice of $\mathbf{w}_{-i}^{(1)}$ owes same objective value. Thus, from Eq. (17), we still obtain $\ell^T X_{-i} = \mathbf{0}$. Thus any local minimum is a global minimum for the pruned sparse model. $\square$

### F.2 PROOF OF THEOREM 7

Proof: Since $\forall\, i \neq j$, $X_{-i}$, $X_{-j}$ share no same columns and $X^T X = I_d$, then $\forall\, i \neq j$, $X_{-i}^T X_{-j} = \mathbf{0}$. Besides, from our assumption $\cap_{j=1}^m \mathcal{S}_j = \emptyset$, then $\cap_{j=1}^m \mathcal{S}_j = \{1, \ldots, d\}$, meaning that $(X_{-1}, \ldots, X_{-d})$ is $X$ with different arrangement of columns. Hence

$$Y = \mathcal{P}_X Y + (I - \mathcal{P}_X)Y = X(X^T X)^{-1} X^T Y + (I - \mathcal{P}_X)Y \triangleq \sum_{i=1}^d X_{-i} Z_i + (I - \mathcal{P}_X)Y, \quad (18)$$

Set $W^{(2)} = [\boldsymbol{a}_1, \ldots, \boldsymbol{a}_{d_1}]^T$, then the objective becomes

$$\frac{1}{2} \sum_{i=1}^{d_1} \| Z_i - \mathbf{w}_{-i} \boldsymbol{a}_i^T W^{(3)} \cdots W^{(L)} \|_F^2.$$

We set $\widetilde{X} = X \widetilde{W}^{(1)} = (X_{-1} \mathbf{w}_{-1}, \ldots, X_{-d_1} \mathbf{w}_{-d_1})$. Now we show the following problems have same local minimizer condition for $\mathbf{w}_{-1}$.

$$\textbf{(P)} \quad L(\widetilde{W}^{(1)}, W^{(2)}, \ldots, W^{(L)}) = \frac{1}{2} \left\| Y - \widetilde{X} \left( W^{(2)} \cdots W^{(L)} \right) \right\|_F^2,$$

$$\textbf{(P1)} \quad L_2(\widetilde{W}, A) = \frac{1}{2} \left\| Y - \sum_{i=1}^{d_1} X_{-i} \mathbf{w}_{-i} \boldsymbol{a}_i^T \right\|_F^2. \quad (19)$$

If there is a local minimizer $\mathbf{w}_{-1}, \ldots, \mathbf{w}_{-d_1} \neq \mathbf{0}$, for problem (P), since $d_i \geq \min\{d_1, d_y\}, \forall i \geq 1$ and $\widetilde{X}, Y$ have full column rank, then based on Theorem 2.3 in Lu & Kawaguchi (2017), a local minimizer of $L(\widetilde{W}^{(1)}, W^{(2)}, \ldots, W^{(L)})$ is obtained when

$$W^{(2)} \cdots W^{(L)} = \left( \widetilde{X}^T \widetilde{X} \right)^{-1} \widetilde{X}^T Y.$$

Notice that $\widetilde{X}^T \widetilde{X} = \mathrm{diag}(\mathbf{w}_{-1}^T \mathbf{w}_{-1}, \ldots, \mathbf{w}_{-d_1}^T \mathbf{w}_{-d_1})$. Then the objective is simplified as

$$2L(\widetilde{W}^{(1)}) = \left\| Y - \widetilde{X} \left( \widetilde{X}^T \widetilde{X} \right)^{-1} \widetilde{X}^T Y \right\|_F^2 = \left\| Y - \sum_{i=1}^{d_1} \frac{(X_{-i} \mathbf{w}_{-i})(X_{-i} \mathbf{w}_{-i})^T Y}{\mathbf{w}_{-i}^T \mathbf{w}_{-i}} \right\|_F^2$$

$$= \sum_{i=1}^{d_1} \left\| X_{-i} Z_i - \frac{(X_{-i} \mathbf{w}_{-i})(X_{-i} \mathbf{w}_{-i})^T X_{-i} Z_i}{\mathbf{w}_{-i}^T \mathbf{w}_{-i}} \right\|_F^2 \quad (20)$$

$$= \sum_{i=1}^{d_1} \left\| X_{-i} Z_i - \frac{X_{-i} \mathbf{w}_{-i}^T \mathbf{w}_{-i} Z_i}{\mathbf{w}_{-i}^T \mathbf{w}_{-i}} \right\|_F^2$$

For problem (P1), similarly, a local minimizer of $L_2(\widetilde{W}, A)$ is obtained when $(X_{-j} \mathbf{w}_{-j})^T \left( Y - \sum_{i=1}^{d_1} X_{-i} \mathbf{w}_{-i} \boldsymbol{a}_i^T \right) = 0$. Then $\boldsymbol{a}_j^T = \frac{(X_{-j} \mathbf{w}_{-j})^T Y}{\mathbf{w}_{-j}^T \mathbf{w}_{-j}}$, showing same loss objective as

$$2L_2(\widetilde{W}) = \left\| Y - \sum_{i=1}^{d_1} \frac{(X_{-i} \mathbf{w}_{-i})(X_{-i} \mathbf{w}_{-i})^T Y}{\mathbf{w}_{-i}^T \mathbf{w}_{-i}} \right\|_F^2$$

$$= \sum_{i=1}^{d_1} \left\| X_{-i} Z_i - \frac{(X_{-i} \mathbf{w}_{-i})(X_{-i} \mathbf{w}_{-i})^T X_{-i} Z_i}{\mathbf{w}_{-i}^T \mathbf{w}_{-i}} \right\|_F^2 = 2L(\widetilde{W}^{(1)}). \quad (21)$$

Finally, based on Theorem 2, every local minimum of (P1) is a global minimum. Hence every local minimum of (P) is a global minimum.

If there exists $i_0$, such that $\mathbf{w}_{-i_0} = \mathbf{0}$, we show that $Z_{i_0} = \mathbf{0}$ below. Without loss of generality, we assume $i_0 = 1$, then the value of $\boldsymbol{a}_1$ does not affect the objective, we take $\boldsymbol{a}_1 = \mathbf{0}$ as well. In order

to show the result, we only perturb $\mathbf{w}_{-1}, \boldsymbol{a}_1, W^{(3)}, \ldots, W^{(L)}$ into $\mathbf{w}_{-1} + \Delta\mathbf{w}, \boldsymbol{a}_1 + \Delta\boldsymbol{a}, W^{(3)} + \Delta_3, \ldots, W^{(L)} + \Delta_L$ and analyze the difference of loss as $\Delta L$. We set

$$\Delta W \triangleq \prod_{i=3}^{L} \left(W^{(i)} + \Delta_i\right) - \prod_{i=3}^{L} W^{(i)}, \ W^o \triangleq \prod_{i=3}^{L} W^{(i)}. \tag{22}$$

Then the perturbation leads to

$$\begin{aligned}
2\Delta L(\Delta\mathbf{w}, \Delta\boldsymbol{a}, \Delta W) &= \|Z_1 - \Delta\mathbf{w}\Delta\boldsymbol{a}^T \left(W^o + \Delta W\right)\|_F^2 - \|Z_1\|_F^2 \\
&\quad + \sum_{i \neq 1} \|Z_i - \mathbf{w}_{-i}\boldsymbol{a}_i^T \left(W^o + \Delta W\right)\|_F^2 - \|Z_i - \mathbf{w}_{-i}\boldsymbol{a}_i^T W^o\|_F^2 \\
&= -2tr[Z_1^T \Delta\mathbf{w}\Delta\boldsymbol{a}^T \left(W^o + \Delta W\right)] + \|\Delta\mathbf{w}\Delta\boldsymbol{a}^T \left(W^o + \Delta W\right)\|_F^2 \\
&\quad - 2\sum_{i \neq 1} tr[\Delta W^T \boldsymbol{a}_i \mathbf{w}_{-i}^T \left(Z_i - \mathbf{w}_{-i}\boldsymbol{a}_i^T W^o\right)] + \|\mathbf{w}_{-i}\boldsymbol{a}_i^T \Delta W\|_F^2.
\end{aligned} \tag{23}$$

Applying the first case to the remaining parameters excluding $\mathbf{w}_{-1}$ and $\boldsymbol{a}_1$ (If there are several $\mathbf{w}_{-i}$s are zero, we can leave them all out), we have

$$\boldsymbol{a}_i^T W^o = \frac{(X_{-i}\mathbf{w}_{-i})^T Y}{\mathbf{w}_{-i}^T \mathbf{w}_{-i}} = \frac{(\mathbf{w}_{-i})^T Z_i}{\mathbf{w}_{-i}^T \mathbf{w}_{-i}},$$

which agrees with

$$\mathbf{w}_{-i}^T \left(Z_i - \mathbf{w}_{-i}\boldsymbol{a}_i^T W^o\right) = \mathbf{0}, i \neq 1.$$

Hence the second term in the final row of Eq. (23) is zero. Besides, let us note the first-order term of $\Delta\mathbf{w}$, showing that $tr[Z_1^T \Delta\mathbf{w}\Delta\boldsymbol{a}^T \left(W^o + \Delta W\right)] = 0$. Otherwise, given $\mathbf{w}_{-1} = \Theta(t^{-1}), \boldsymbol{a}_{-1} = \Theta(t^{-1}), \Delta W = \Theta(t^{-3})$, as $t \to \infty$, the sign in the final expansions of Eq. (23) depends on the fist term that is indefinite.

Therefore, $\Delta\boldsymbol{a} \left(W^o + \Delta W\right) Z_1^T = \mathbf{0}$, then $\left(W^o + \Delta W\right) Z_1^T = \mathbf{0}$, leading to $W^o Z_1^T = \mathbf{0}$ and $\Delta W Z_1^T = \mathbf{0}$.

In view of expression $\Delta W$, it holds that

$$\begin{aligned}
\Delta W Z_1^T &= \sum_{i=3}^{d_1} \left(W^{(3)} \cdots W^{(i-1)} \Delta_i W^{(i+1)} \cdots W^{(L)}\right) Z_1^T + \ldots \\
&= \sum_{t=1}^{L-2} f_t(\Delta_3, \ldots, \Delta_L) Z_1^T,
\end{aligned} \tag{24}$$

where $f_t(\Delta_3, \ldots, \Delta_L)$ is the sum of the product in $W^{(3)}, \ldots, W^{(L)}, \Delta_3, \ldots, \Delta_L$ that contains exactly $t$ different $\Delta_i$s. Then from small-order terms to high-order terms, we obtain $f_t(\Delta_3, \ldots, \Delta_L)Z_1 = \mathbf{0}$. It follows from $f_{L-2} = \Delta_3 \cdots \Delta_L$, $d_i \geq \min\{d_1, d_L\}$, and the arbitrary of $\Delta_3 \cdots \Delta_L$, we get $Z_i = \mathbf{0}$. Finally, when $Z_1 = \mathbf{0}$, It is evident that $\mathbf{w}_1 = \mathbf{0}$ already satisfies the global minimizer condition since the objective is separated as $\sum_{i=1}^{d_1} \|Z_i - \mathbf{w}_{-i}\boldsymbol{a}_i^T W^{(3)} \cdots W^{(L)}\|_F^2$. This completes the proof.

$\square$

