# OpenReview forum: "On the Landscape of Sparse Linear Networks"
_ICLR.cc/2021/Conference — Reject_

### Official Review · AnonReviewer1 · 2020-10-20
**Understanding the landscape of sparse linear networks is a worthwhile goal, but this submission does not offer much insight**

**Rating:** 4
**Confidence:** 4

**Review:**

This paper studies the optimization landscape of (deep) sparse linear networks. The study of sparse neural networks is well motivated: on the one hand, there is a lot of experimental evidence that the loss of the trained network does not decrease much after removing a large subset of the connections; on the other hand, there is little theoretical evidence of what makes this behaviour possible (or how to provably construct sparse networks from dense ones). As a result, an investigation of the optimization landscape of sparse networks, even in the simple case of a linear activation function, is timely and interesting to the ICLR community, since it can potentially shed light on the questions above.

The main contributions of this paper can be summarised as follows:

(1) If the network has a scalar output, then every local minimum is global (Theorem 1 for two-layers and Theorem 6 for multiple layers).

(2) The same conclusion (every local min is global) holds if the network has two outputs under extra assumptions (non-overlapping first layer weights and orthogonal data).

(3) If the network has at least three outputs, it is possible to build counterexamples with spurious local minima (Theorem 5 gives one such counterexample and the algorithm in Section 3 gives even more of them).

Overall, although the study of the landscape of sparse linear networks is an interesting topic, the submission is not particularly insightful in this regard. The results are quite weak and only show that spurious local minima can occur as soon as the output dimension is 3 (while they provably do not occur in lower output dimensions). The positive results for two-layer networks follow from a straightforward analysis and they can be extended to multiple layers as done in the related literature (Lu & Kawaguchi, 2017; Eftekhari, 2020). The negative results consist in presenting a counterexample.

The main weaknesses of the submission are as follows:

(1) The paper does not really explain why neural networks can be pruned so well. It just proves that the landscape can be bad.

(2) The analysis is quite straightforward and does not enough elements of novelty over existing work.

One way to obtain more insightful results would be to consider a less general setting. In the current version, the authors prove that spurious local minima exists for some initialization and for some data distribution. Does the same conclusion hold for "most" data distribution (say a Gaussian input distribution) or "most" initializations (say Xavier's initialization)? Proving such a statement may lead to significantly more interesting results.

---

### Official Review · AnonReviewer4 · 2020-10-26
**Interesting new results on deep linear networks**

**Rating:** 7
**Confidence:** 5

**Review:**

This paper provides an interesting analysis and direction to understand landscape of deep linear networks with sparsity patterns. In terms of main technical contributions, the authors prove that every local minimum is a global minimum in the case of scalar output with any sparsity structure, and show the existence of sub-optimal local minima in the case of high-dimensional output, analytically by considering a simple shallow network with a simple sparsity structure, and numerically by investigating the solutions found by an algorithm that converges to a local minimum.

Understanding deep linear networks is an important topic for several reasons. One reason is that deep linear networks change its NTK during training, unlike the unrealistic deep nonlinear networks in the regime of extreme over-parameterization. Deep nonlinear networks in the regime of extreme over-parameterization have been studied in the regime of extreme over-parameterization, or so-called NTK regime. However, we know now that deep nonlinear networks in the regime of extreme over-parameterization is indeed shallow linear models with a fixed kernel, or a fixed NTK. In other words, analyzing deep nonlinear networks in the regime of extreme over-parameterization is reduced to analyze classical shallow linear models, which is trivial in terms of theory. Moreover, from practical viewpoint, we know that practical deep neural networks change NTK during training, which seems to be one of the main sources of inductive bias to allow deep networks to perform well. To understand neural network in the practical regime beyond the NTK regime, for example, deep linear network with one hidden layer was recently studied in https://arxiv.org/abs/2003.02218.

In sum, understanding deep linear networks is currently revisited and of the great importance in the literature, as we now realize that deep nonlinear networks in extreme over-parametrization regime or NTK regime are simply shallow linear models. Given this context, I think this paper provides an interesting and important contribution to the literature.

The proof of Theorem 5 makes sense. Due to the sparsity pattern, we cannot escape from w_4 = 2_8 = 0 without increasing the loss value in the sufficiently small neighborhood.

On the other hand, the particular construction used in the proof of Theorem 5 is not general in the following sense. Given this target matrix Y and this sparse network, w_3 and w_7 are unnecessary, and we know a priori that we should make w_3 and w_7 to be zero. Indeed, the existence of the sub-optimal local minima is shown by relying on the nonzero w_3 and w_7 in the proof. An interesting question to further improve the paper would be the following: can we construct suboptimal local minima with w_3 = w_7 = 0 in this example? With w_3 = w_7 = 0, the perturbations of w_4 and w_8 can always decrease the loss value by making w_4 * w_8 closer to 4.


Minor comments:

- if the term “sub-network” is used in the statement of the theorems, it should be more formally defined somewhere. I could understand what it means by reading the proof.

---

### Official Review · AnonReviewer3 · 2020-10-28
**Lack novelty and significance**

**Rating:** 4
**Confidence:** 4

**Review:**

This paper studies the loss landscapes of sparse linear networks. It proves that under squared loss, (1) spurious local minimum does not exist when the output dimension is one, or with separated first layer and orthogonal training data; and (2) for two-layer sparse linear networks, the good property in (1) does not exist anymore when the conditions are violated. The authors also report experimental results to show that two-layer sparse linear networks with two hidden neurons have spurious local minima.

Overall, I vote for rejection. The proofs are detailed and seem correct. However, I was worried that (1) the proofs make incremental contribution compared with two existing works; (2) assuming activations are linear is too strict; (3) the insight given by the theorems is not clear; and (4) the applicable domain is not clear.

Pros:

+ The authors prove that (1) sparse linear networks do not have any spurious local minima under some assumptions; and (2) for two-layer sparse linear networks, the previous properties do not stand anymore when the assumptions no longer hold. The authors have given detailed proofs for their arguments.

Cons: I was concerned about the technical novelty and significance. They are not justified in this paper.

- From technical aspects, the proves are simple extensions of existing works on linear neural networks; such as Kawaguchi (2016) and Lu & Kawaguchi (2017). The authors do not state clearly what new proof techniques are used in this paper.

- This paper assumes all activations are linear and then proves in many cases spurious local minima do not exist. However, most neural networks in practice _have_ nonlinear activations. Some papers have suggested the nonlinearity in activations significantly change the loss landscape, such as [1-3] amongst others. I understand that it would be good to study a simpler model when a comprehensive model is intractable. But the simpler model needs to share similar properties with the real-world case. The three aforementioned papers concern me.

- The insights of the given theories are not clear. Did the authors hope to suggest pruning would bring something new into linear networks? It would be good if the authors can clearly discuss this and give some potential practical implications.

- The potential application would be limited. Understanding the loss landscapes of sparse linear networks is important if we need to train them from scratch. However, most pruning methods are applied when neural networks have been trained. Some works even do not need fine-tuning after pruning. This undermines the importance of this paper.

Questions: It would be good if the authors can comment on the cons.

[1] Yun, C., Sra, S., and Jadbabaie, A., “Small nonlinearities in activation functions create bad local minima in neural networks,” ICLR 2019.
[2] He, F., Wang, B., and Tao, D., “Piecewise linear activations substantially shape the loss surfaces of neural networks,” ICLR 2020.
[3] Goldblum, M., Geiping, J., Schwarzschild, A., Moeller, M., and Goldstein, T., “Truth or backpropaganda? An empirical investigation of deep learning theory,” ICLR 2020.

---

### Official Review · AnonReviewer2 · 2020-10-28

**Rating:** 5
**Confidence:** 4

**Review:**

Summary

This paper considers the optimization landscape of sparse linear networks. Under the regression setup (squared error loss), the paper shows that there is no spurious local minimum in the training loss (as in the dense linear networks) if:
1) The output label is scalar ($d_y = 1$), or
2) The data matrix $X \in \mathbb R^{n \times d_x}$ has orthogonal columns, only the first layer of the network is sparse, and each input node is connected to only one hidden node in the first hidden layer (similar to non-overlapping filters in convnets).

If these conditions are not met, there may be bad local minima caused by the sparsity of the network. The paper gives some examples:
1) For $d_y = 3$, the paper gives a counterexample sparse network that has a bad local minimum.
2) For $d_y = 2$ (and higher), the paper proposes an iterative algorithm that looks for local minima of the sparse neural network, and it verifies by experiments that bad local minima exist.

Strength

Although sparse neural networks and network pruning has received much attention, our understanding of sparse models from the optimization point of view is limited. It is good to have some results in this field. I value the theoretical investigations done in this paper, and I’m also a bit surprised that there is such a stark contrast between the cases $d_y = 1$ and $d_y > 1$. The paper essentially tells us that sparse linear networks have benign loss landscapes as in the dense linear networks if the output is scalar (Theorem 1 & 6), but suffer bad local minima when $d_y > 1$ unless some strong conditions are satisfied.

Weakness

Although I find the observations in this paper interesting, I cannot stop questioning whether the analysis done in this paper is really meaningful or useful. It is known that there is already a big gap between linear networks (i.e., without nonlinear activation functions) vs nonlinear networks (i.e., with nonlinear activations), in terms of the “no bad local minima” property. Linear networks (under some width conditions) have no bad local minima, but nonlinear networks do have bad local minima. Intuitively speaking, sparsifying network weights will “hurt” the loss landscape. Given that nonlinear networks already have bad local minima even before sparsifying, it is natural to expect that sparse nonlinear networks will also have bad local minima. My question is, what kind of insight does the paper provide for nonlinear networks, other than the (already expected) conclusion that “sparse nonlinear networks likely have bad local minima”? In my humble opinion, the significance of the analysis in this paper looks rather limited and it does not provide new insights that carry over to sparse nonlinear networks.

In addition, the clarity of the paper requires improvement. Before reading the paper, I found it very difficult to parse the abstract, especially the conditions (e.g., “non-intersected sparse first layer and dense other layers with orthogonal training data”) in 1) and 2). It would be better to elaborate more on these conditions. Also, I found the results in Section 2.4 difficult to understand; I think the statements of Theorems 3 and 4 are not clear enough. The terms “some projected data”, “influence for spurious local minima” are not defined nor clear. What is “the same local minimum” in Theorem 4 referring to? A more high-level question is: how are these results related to the existence of bad local minima? In Theorem 7, I couldn’t find the dimensions $d_i$ defined anywhere in the main text.

Although it is claimed in the abstract that “no unrealistic assumptions” are made, I believe the conditions in Theorems 2 and 7 are quite strong. The theorems require that a) the data matrix $X$ has orthogonal columns, b) only the first layer of the network is sparse, and c) each input node is connected to only one hidden node in the first hidden layer. I believe these conditions are not likely to arise in practice. It’d be great if you can provide more justifications on these set of conditions.

Overall assessment

Although the paper makes some contributions to the understanding of the sparse linear networks, I believe the significance of the results are rather limited and they do not provide additional insights. I lean towards rejection.

Minor comments
- The paper is structured in a way that all results on shallow networks are presented first and then some of them are extended to deep networks. However I had the false impression that all the results only hold for shallow networks before reaching Section 4, which is at the end of the paper. I think it’d be good to have a remark after Theorems 1 and 2 that they are extended to deep networks in a later section.
- In Section 3, is it difficult to construct some counterexamples ($d_y = 2$) with bad local minima, instead of using an iterative algorithm to find them? It’d be good if you can show that the conditions in Theorems 2 and 7 are necessary.
- Fig 2: typos in hidden node superscripts?
- Pg 7, first line: Eq. (5) -> Eq. (7)?

---

### Decision · Program_Chairs · 2021-01-07
**Final Decision**

**Decision:**

Reject

**Comment:**

The paper studies optimization landscapes arising the fitting of sparse linear networks to data. It argues that for scalar outputs, every local minimum is global, while for d >= 3 dimensional outputs, there can be spurious local minimizers. The paper also argues that similar results hold for deep networks. Counterexamples on the existence of non-global local minimizers are constructed analytically and corroborated by probing the optimization landscape experimentally.

Pros and cons:

[+] Network sparsification is an important practical problem, and there are relatively few theoretical guidelines on when and how sparsification can be achieved. In particular, results that help to explain why trained networks are often sparsifiable and/or provide theoretical guarantees for sparsification algorithms would be significant.

[-] Reviewers raised concerns about the significance of the paper’s results. In particular, they found it difficult to connect the paper’s analysis of landscapes of linear networks to the question of when and why practically occurring networks can be pruned. They also had difficulty isolating new ideas in the mathematical analysis beyond previous works on the landscape of linear networks. Finally, several reviewers expressed concerns about the extent to which the paper’s observations on linear networks generalize to nonlinear networks.

[-] Reviewers raised concerns about the clarity of the paper. The mathematical exposition is unclear in places: conditions are not clearly stated, terminology is occasionally vague. Moreover the paper’s handling of optimality conditions for constrained optimization problems is unclear (e.g., the proof of Theorem 6 uses the unconstrained optimality condition $\ell^T X = 0$ in the inductive step, even though the inductive hypothesis pertains to a constrained problem).

[-] The technical results make assumptions that are occasionally quite strong. For example, Theorem 7 requires orthogonality of the data matrices, when restricted to indices where the weights are nonzero. As reviewers note, this assumption seems highly restrictive.

Overall, the paper addresses a topic that is important to the ICLR community: developing theoretical analyses of network sparsification. However, the significance of its results is not clear, and the technical exposition would need significant improvement to meet the bar for publication.